# The Impact of COVID-19 Vaccination on Anxiety Levels of Turkish Dental Professionals and Their Attitude in Clinical Care: A Cross-Sectional Study

**DOI:** 10.3390/ijerph181910373

**Published:** 2021-10-01

**Authors:** Fatih Karayürek, Ahmet Taylan Çebi, Aydın Gülses, Mustafa Ayna

**Affiliations:** 1Department of Periodontology, Cankiri Karatekin University, Cankiri 18200, Turkey; 2Department of Oral and Maxillofacial Surgery Karabük University, Karabük 78050, Turkey; ahmettaylancebi@karabuk.edu.tr; 3Department of Oral and Maxillofacial Surgery, Universitätsklinikum Schleswig Holstein, Campus Kiel, Christian Albrechts University, 24105 Kiel, Germany; aguelses@mkg.uni-kiel.de; 4Department of Periodontology, Bonn University, 53111 Bonn, Germany; praxis@dr-ayna.de

**Keywords:** anxiety, dental, pandemic, profession, equipment

## Abstract

Background: The current study aimed to assess the anxiety and fear levels and the attitude towards clinical care, such as the use of personal protective equipment and number of patients examined, before and after COVID-19 vaccination among Turkish dental professionals. Methods: A questionnaire including socio-demographical characteristics and clinical data regarding the number of patients, the use of personal protective equipment, vaccine confidence interval, positive or negative COVID-19 diagnosis, and fear and anxiety levels were examined. Results: A total of 475 dentists (196 men and 279 women) participated. Overall, the vaccination had a positive effect on the decrease of fear and anxiety levels of dental professionals. It was observed that the number of interventional procedures significantly increased after vaccination. Besides that, the amount of personal protective equipment used in patients, especially after the vaccination, has decreased. Conclusion: Despite the positive effects of vaccination on the anxiety levels of dental professionals, protective measurements should further be the main concern, regardless of the vaccination status of both the dental professional and the patient.

## 1. Introduction

The infection known as severe acute respiratory syndrome coronavirus (SARS-CoV-2) and named COVID-19 by the World Health Organization (WHO) was first reported in China in December 2019. In February 2020, this viral infection widely spread in most countries and became pandemic that created a major global public health crisis.

Turkey successfully tackled this outbreak and kept it under control. Nevertheless, the number of cases showed increases in the following days of the pandemic. On 16 April 2021, the number of daily cases reached 63,000, and this number was a new summit [1]. As a consequence of measures taken nationally, intensive care and ventilated patients decreased. However, in the new normalization process, there have been increases in new cases and the number of critically ill patients, as a result of disregarding the mask, distance, and hygiene rules and increased crowding. According to the data of the Turkish Ministry of Health, a new summit was reached in the number of critically ill patients on 15 December 2020 (5988 patients) [1].

Towards the end of 2020, with the announcement of vaccines produced by state-sponsored and private companies after hard work [2] and approval by the WHO, new cases, the number of critically ill patients, and deaths dramatically decreased with mass immunization campaigns either globally or nationally. Vaccines prevent the spreading of contagious diseases and are a crucial factor for decreasing deaths. The vaccine hesitancy also attracted the attention of the WHO, declaring it as one of the biggest threats to global health [3]. Among the basis of hesitancy in society towards the vaccine are factors such as religious beliefs, lack of confidence in healthcare professionals, certain personality disorders, and financial situations [4]. It has been stated that it is important to act jointly to increase the public’s confidence in the light of scientific data in the field of health workers to reduce hesitancy about vaccination [5]. In the study of Napolitano et al., it was stated that lack of knowledge was an important factor in the high rate of vaccine hesitancy. According to the study, more than half of parents stated that they lacked knowledge on childhood immunizations [6]. It is important to reach a reliable source of information to reduce vaccine hesitancy. Dubé et al. showed that vaccine hesitancy increased with information obtained from the internet and social media [7].

Because the most common transmission of COVID-19 infection is through the respiratory system, and a large amount of aerosol is produced during these procedures, dentistry is considered among the riskiest professions [2,8]. Moreover, dental procedures facilitate the spread of infectious diseases to dentists or other patients due to close contact if preventive measures are not taken [9]. Since dentists generally use personal protective equipment (PPE) during dental treatments, they have easily adapted to the conditions of the COVID-19 pandemic, as they determine the possible risk of cross-infection that may result from infectious diseases that may occur in the examination conditions and take precautions accordingly. After the announcement of the COVID-19 pandemic, there has been uncertainly about which PPE is most effective. These protective measures varied globally and nationally. The deficiency of evidence-based research of proposed guidelines caused this situation [10].

The present study aimed to compare changes in the usage of PPE, attitudes, behaviors, anxiety, and fear levels before and after COVID-19 vaccination with the hypothesis that the vaccination could have possible positive effects on the attitude of dental practitioners and increase the number of patients treated daily.

## 2. Materials and Methods

### 2.1. Study Design

This study was conducted with the participation of dental professionals who actively worked at public institutions, private clinics, and university clinics. The study was embarked upon by receiving approval from The Ethics Committee of non-interventional Clinical Research (2 June 2021, 2021/572) and was conducted according to the guidelines of the Helsinki Declaration of Human Rights. All participants were informed, and a “voluntary informed consent form” was obtained from those who accepted to participate in this study. The “voluntary informed consent form” section was added before the questionnaire section, and they were required to read and approve the “voluntary informed consent form” before answering the online questionnaire. Participation of individuals in the survey was based on voluntariness.

### 2.2. Data Collection

In the selections of the dentists participating in this study, metropolitan cities with higher COVID-19 cases were selected according to the data of the Ministry of Health. The online questionnaire links were sent to the personal e-mails of dentists working in universities, public institutions, and private clinics, and dentists who agreed to participate in the study were included in the study. Moreover, the questionnaire forms were sent through WhatsApp Messenger (WhatsApp Inc., Menlo Park, CA, USA), Facebook (Facebook, Inc., Menlo Park, CA, USA), and Instagram (Menlo Park, CA, USA). The surveys were prepared on “Google Forms”, and the survey link was sent to the participants. The data obtained from the participants were collected anonymously in the “Google Forms” dataset.

A 22-question questionnaire was directed to participates. Socio-demographic (7 questions) and questions about the COVID-19 outbreak (15 questions) constituted the questionnaire. After sociodemographic questions regarding factors such as age, gender, and working life, for before and after vaccination periods, questions were asked about the number of patients, PPE, vaccine confidence interval, positive or negative COVID-19 diagnosis, and fear and anxiety levels that may likely occur due to the busy working hours of the dentists. Participants who answered the survey questions were obliged to answer all questions (Table 1).

### 2.3. Statistical Analyses

Study data were analyzed using “Python” (open-source programming language) and SPSS (version 28.0.0.0., Chicago, IL, USA). At first, means and standard deviations were calculated. The Shapiro–Wilk test was performed to assess the distribution of the parameters. Chi-squared, ANOVA, Tukey’s tests, logistic regression analysis, McNemar’s test, and Wilcoxon Rank test were applied. Cronbach’s alpha was used to clarify the reliability of the questions. The level of significance was set at *p* < 0.05.

## 3. Results

### 3.1. Socio-Demographic Results

According to data from the Turkish Dental Association, the number of dentists working in Turkey was 39,438 dentists in 2020 [11]. On the basis of a 95% confidence interval, the required sample size was 381. On the basis of the number of 381, the questionnaires reached 639 dentists living in various metropolitan areas, but only 475 dentists were approved to participate in the study (74.3%).

The mean age of participants was 36.42 ± 7.37 with an age range between 25 and 65 years, and average professional experience duration was also calculated as 11.8 ± 8.01 years. The mean reaction time to participation was 46.8 h. A total of 146 of participants (30.7%) who were included in this study were general dentists, and specialist dentists consisted of 83 oral surgeons, 40 oral and maxillofacial radiology specialists, 51 pediatric dentists, 87 periodontists, 67 prosthodontists, and only 1 restorative dentistry specialist. A total of 174 (36.6%) of participants served at public institutions. For private clinics and universities, these numbers were 133 (28%) and 168 dentists (35.4%), respectively. The years of professional experience were as follows: 38.7% between 0 and 10 (*n* = 184), 40.8% between 11 and 20 (n = 194), and 20.4% over 21 years of experience (*n* = 97) (Table 2). Exactly 48 participants (10.1%) remarked that they had been infected by systemic diseases, while eight participants (1.7%) stated that they had been not vaccinated; 96 of vaccinated participants (20.5%) remarked that antibody measurement was not performed.

### 3.2. The Data of COVID-19 Outbreak Questions

Primarily, Cronbach’s alpha value was calculated to measure the reliability and consistency of the questions asked to the participants in the questionnaire (0.718). Logistic regression analyses were used to indicate how the independent variables (institutions, specialty, experience duration, etc.) affect the dependent variable. Moreover, McNemar’s test was applied to evaluate the alterations that occurred before and after the COVID-19 vaccine.

As part of the COVID-19 vaccination program in Turkey, it is known that only CoronaVac (Sinovac Life Sciences, Beijing, China) was administrated to dentists who ranked in the priority group in January–March 2021. The confidence interval value known as CoronaVac was asked of participants, and according to answers of all participants, this value was determined as 5.97 out of 10. The number of dentists who caught COVID-19 before the vaccination was 71 (14.9%), whereas 32 participants (6.7%) caught COVID-19 after the vaccination. This alteration showed a statistically significant difference (*p* = 0.001) (Table 3).

The number of COVID (+) diagnoses of dentists working in public institutions was 29 before vaccination but was 17 after vaccination. This decrease did not show a statistically significant difference (*p* = 0.09). For before and after the vaccination, the number of dentists working in universities was 21 and 8, respectively; for private clinics, this was 21 and 7, respectively. These decreases were significantly different in both groups (*p* = 0.011, *p* = 0.026, respectively). The number of COVID (+) diagnoses for specialist dentists was 51 and 21 before and after vaccination, respectively; for general dentists, it was 20 and 11, respectively. While the alterations in specialist dentists before and after vaccination showed statistically significant differences (*p* = 0.001), the decreases in the group of general dentists did not show statistically significant differences (*p* = 0.137). The number of COVID (+) diagnoses of dentists having between 0 and 10 years professional experience were 18 and 16 in the pre-and post-vaccination periods, respectively (*p* = 0.864). We found that 29 and 9 dentists with between 11 and 20 years professional experience, and 24 and 7 dentists with over 21 years of experience reported diagnosis of COVID (+), respectively (*p* = 0.002, *p* = 0.002, respectively) (Table 4).

The dentists participating in this study were asked about PPE used by dentists in patients with whom they performed dental procedures before and after vaccination, and it was seen that there were decreases in the amount of PPE used in patients, especially after having had the vaccination. While the average amount of PPE used before the vaccination was 4.69, this number descended to 4.32 after the vaccination. There was a 7.7% decrease in PPE use after vaccination. This decrease showed a statistically significant difference (*p* = 0.001) (Table 3). Before their vaccination, 396 participants stated that the number of patients they examined during the outbreak was between 0 and 10, while after their vaccination, the number of participants who received these patients between 0 and 10 decreased to 366. While the number of participants who stated that they examined 11 to 20 patients before their vaccination was 49, this number was 71 after vaccination. The number of participants who examined 21 or more patients was 30 and 38 before and after vaccination, respectively. It was observed that the number of patients received by all dentists increased after vaccination, and these increases showed statistically significant differences (*p* = 0.039) (Table 3).

The average of PPE used by dentists working in public institutions was 4.79 before vaccination but was 4.54 after vaccination. This decrease did not show a statistically significant difference (*p* = 0.088). For before and after vaccination, the average amount of PPE used by dentists working in universities was 4.66 and 4.24, respectively; for private clinics, this was 4.6 and 4.15, respectively. These decreases were statistically significantly different in both groups (*p* = 0.008, *p* = 0.006, respectively). The average amount of PPE used by specialist dentists was 4.7 and 4.24 before and after vaccination, respectively; for general dentists, this was 4.67 and 4.52, respectively. While the decreases in the mean PPE numbers of specialist dentists before and after vaccination showed statistically significant differences (*p* = 0.001), the decreases in the amount of PPE in general dentists were not statistically significantly different (*p* = 0.323). Likewise, the average amount of PPE use was examined in terms of experience duration for before vaccination, and the average was 4.52 in dentists having between 0 and 10 years of professional experience, 4.9 for dentists having between 11 and 20 years professional experience, and 4.59 for those with over 21 years of experience. After vaccination, the averages were 4.21, 4.56, and 4.06, respectively. Statistically significant differences were observed in the groups of three different durations of experience (*p* = 0.022, *p* = 0.022, *p* = 0.012, respectively) (Table 4).

While the confidence value for the CoronaVac vaccine was 5.88 in dentists working in public institutions, for dentists working in private clinics and dentists working in universities, the numbers were 6 and 6.05, respectively. There were no significant differences between institutions and the confidence interval value (*p* = 0.763) (Table 5). According to the answers given to the questions asked about the occurrence of anxiety and fear problems experienced by dentists during the COVID-19 pandemic, 134 of the participants working in public institutions (77%), 103 of the participants working in private clinics (77.4%), and 127 of the dentists working in universities (75.6%) stated that they had anxiety and fear problems due to the COVID-19 pandemic. It was observed that there was no significant difference in the data obtained according to the institution (*p* = 0.462). A total of 64 of the participants working in public institutions (39.6%) stated a decrease in anxiety and fear after the vaccination. These percentages for dentists working in private clinics (*n* = 39) and universities (*n* = 61) were 29.3% and 36.3%, respectively. There were no statistically significant differences between the institutions and the decreases of anxiety and fear dependent on the COVID-19 pandemic (*p* = 0.43). The percentages of dentists considering leaving their duties due to the COVID-19 pandemic in the period before vaccination for public institutions, private clinics, and universities were 24.1%, 18%, and 14.3%, respectively. These rates were statistically significant differences (*p* = 0.024) in the number of dentists who were considering leaving their duties and working in institutions (Table 5).

Before vaccination, 145, 115, and 136 dentists working in public institutions, private clinics, and universities, respectively, stated that the number of patients they examined during the outbreak was between 0 and 10. After vaccination, these numbers decreased to 134, 106, and 126, respectively. The number of participants who stated that they examined 11 to 20 patients before vaccination was 19, 10, and 20, and for post-vaccination, these numbers were 25, 18, and 27, respectively. The number of participants who examined 21 or more patients was 10, 8, and 12, respectively, for pre-vaccination. After vaccination, these numbers increased to 15, 8, and 15, respectively. There were no significant differences between the number of patients examined by dentists working in three different institutions before and after vaccination (*p* = 0.16, *p* = 0.354, and *p* = 0.216, respectively) (Table 4). While there was no significant difference in the inter-institutional comparison of the number of patients examined before vaccination (*p* = 0.627), there was also no significant difference in the inter-institutional comparison of the number of patients admitted after vaccination (*p* = 0.854) (Table 5).

There were no statically significant differences among specialist and general dentists for CoronaVac confidence value (*p* = 0.414). For specialist and general dentists, these values were 6.03 and 5.85, respectively. According to the answers given to the question regarding leaving institutions between specialist and general dentists due to the anxieties and fears experienced by dentists before vaccination during the COVID-19 pandemic, the percentages of dentists considering leaving their duties due to the COVID-19 pandemic in the period of before vaccination in specialist (*n* = 59) and general dentists (*n* = 32) were 17.9% and 21.9%, respectively. These rates were statistically significant differences (*p* = 0.565) in the number of dentists who were considering leaving their duties and working in institutions. While 240 specialist dentists (72.9%) and 124 general dentists (84.9%) answered “yes” to the question of experiencing fear and anxiety due to the COVID-19 pandemic, a statistically significant difference was observed between these results (*p* = 0.007). However, the number of specialist dentists who stated that these decreases in fears and anxiety levels after vaccination was 120 (36.5%), and the number for general dentists was 49 (33.5%). There was no statistically significant difference between these values (*p* = 0.744) (Table 6).

Before the vaccination, 273 specialty dentists and 123 general dentists stated that the number of patients they examined during the outbreak was between 0 and 10. After vaccination, these numbers decreased to 251 and 115, respectively. While the number of participants who stated that they examined 11 to 20 patients before the vaccination were 35 and 14, respectively, for post-vaccination, these numbers were 51 and 20, respectively. The number of participants who examined 21 or more patients was 21 and 9 for pre-vaccination, respectively. After vaccination, these numbers increased to 27 and 11, respectively. There were no significant differences between the number of patients examined by both specialty and general dentists before and after vaccination (*p* = 0.072, *p* = 0.304, respectively) (Table 4). While there was no significant difference in the specialty status comparison of the number of patients examined before vaccination (*p* = 0.808), there was also no significant difference in the inter-institutional comparison of the number of patients admitted after vaccination (*p* = 0.578) (Table 6).

Between the duration of professional experience and the values of confidence in the vaccine, it was 6.13 for dentists with 0–10 years of experience; this ratio was determined as 5.90 for dentists with 11–20 years of experience, and finally as 5.85 for dentists with 21 years or more of experience. On the basis of the year, while the experience of dentists increased, the value of confidence in the vaccine decreased. This inverse proportionality was not statistically significantly different (*p* = 0.475). In participants with a higher professional experience (21 years and over) (14.4%), 14 dentists stated that they were considering leaving their institutions due to the COVID-19 pandemic before vaccination. This number was 36 in individuals with 0–10 years of experience (19.6%), and 41 dentists with 11–20 years of experience (21.1%) stated that they were considering leaving from institutions. There was no statistical difference between these rates (*p* = 0.534). While 140 dentists with 0–10 years of professional experience (76.1%) answered yes to the question of experiencing fears and anxiety due to the COVID-19 pandemic, the number of dentists with 11–20 years of experience was 158. In dentists with 21 years or more experience, this number was also determined as 66. The fact that this number was lower in dentists with a higher experience was statistically significant different (*p* = 0.029). However, the number of dentists with 0–10 years of experience who stated that these fears decreased after vaccination was 71; the number of dentists with 11–20 years of experience was 74; and, lastly, the number of dentists with 21 years and over experience was 30. A statistically significant difference was found between the obtained data (*p* = 0.046). Before the vaccination, there were no statistically significant differences between examined patients and the experience duration of dentists (*p* = 0.795). However, there was a significant difference after the vaccination (*p* = 0.041). After vaccination, the number of patients treated by less experienced dentists increased more than those treated by more experienced dentists. However, it was determined as 83.1% in dentists with 0–10 years of experience, who look after 0–10 patients daily, 84% in dentists with 11–20 years of experience, and 82.8% in physicians with 21 years or more experience. After vaccination, these rates were 73.9%, 82.8%, and 72.2%, respectively. It was observed that the number of patients who dentists performed interventional procedures daily increased after vaccination (Table 7).

Before the vaccination, 153 dentists with 0–10 years of experience stated that the number of patients they examined during the outbreak was between 0 and 10. A total of 18 dentists with 0–10 years of experience stated that the number of patients they examined during the outbreak was between 11 and 20 patients. It was found that 13 dentists examined 21 or more patients. After vaccination, these numbers were found to be 136, 29, and 19, respectively. These changes showed statistically significant differences in dentists with 0–10 years of experience (*p* = 0.031).

Participants were asked regarding preoperative mouth rinsing to patients in a period of before and after the vaccination, and 256 participants (53.9%) administered mouth rinsing for patients before the vaccination. This rate regressed to 41.9% (199 participants) after the vaccination. This alteration was significantly different (*p* = 0.001) (Table 3). A total of 87 dentists working in public institutions stated that preoperative mouth rinsing was administrated to patients before the vaccination; for dentists working in universities or private clinics, these numbers were 95 and 74, respectively, before vaccination. There were no statistically significant differences regarding these data (*p* = 0.427) (Table 5). A total of 55 dentists working in public institutions also stated that preoperative mouth rinsing was administrated to patients after the vaccination, and for dentists working in universities or private clinics, these rates were 78 and 66, respectively, after vaccination. The obtained data between the rates of preoperative mouth rinsing and institutions were not statistically significantly different (*p* = 0.002) (Table 5). In the intra-institutional comparison of participants in the pre-and post-vaccine period, the decrease in the use of preoperative mouth rinsing after vaccination in dentists working in public institutions was statistically significantly different (*p* = 0.001). However, there was no statistically significant difference in dentists working in universities and private clinics (*p* = 0.068, *p* = 0.366, respectively) (Table 4).

A total of 176 specialist dentists stated that preoperative mouth rinsing was administrated to patients before the vaccination; for general dentists, this number was 80. These numbers were not significantly different (*p* = 0.627) (Table 6). A total of 139 specialist dentists stated that preoperative mouth rinsing was administrated to patients after the vaccination; for general dentists, this rate was 60. These decreases were not significantly different (*p* = 0.972) (Table 6). In the intra-group comparison of specialist dentists and general dentists in the pre- and post-vaccine periods, the decrease in the use of preoperative mouth rinsing after vaccination in both specialist dentists and general dentists showed a statistically significant difference (*p* = 0.003, *p* = 0.016, respectively) (Table 4). The decrease in the use of preoperative mouth rinsing in dentists with 11–20 years experience after vaccination showed a statistically significant difference (*p* = 0.001) (Table 4). Participants in all groups stated that preoperative mouth rinsing was reduced after vaccination.

## 4. Discussion

The current study aimed to evaluate attitudes, behaviors, anxiety, and fear levels of dentists toward SARS-CoV-2 in the period of pre- and post-vaccination. Accordingly, the participants were asked about the number of patients they examined, their use of PPE, and the mouthwashes they administered to patients before and after the COVID-19 vaccine. After vaccination, it was seen that the number of patients received by dentists increased, the equipment they used for personal protection decreased compared to the period at the beginning of the outbreak, and the rate of mouthwash application given to patients preoperatively decreased. In addition, as expected, the number of COVID-19 positives in the participants before the vaccine decreased compared to the post-vaccine period. The decrease in the number of COVID-19 positives in the participants after the COVID-19 vaccine is in line with other studies. In one of these studies, Yassi et al. reported that 3.3% of vaccinated healthcare professionals were infected [12].

Mouthwashes are frequently administered as a preoperative with antiseptic features by dentists [13]. Although it is stated that large-scale clinical studies, including control groups, are needed to measure the effectiveness of antiseptic mouthwashes on SARS-CoV-2, it is recommended that mouthwashes be used to reduce the viral effect of SARS-CoV-2 and to reduce the risk of cross-infection during interventional procedures [14]. In the present study, the rate of recommending antiseptic mouthwash by dentists was 53.9% before the vaccination; this rate regressed to 41.9% after the vaccination. Since the effect of the mass immunization in decreasing COVID-19 transmission risk and the number of intubated patients in intensive care was known [15], in this study, it can be thought that the decrease of the rate of patients’ mouthwash administered by vaccinated dentists compared to in the period before the vaccination is a reflection of the confidence in dentists after vaccination. The percent of mouthwash application to patients decreased in dentists working in all institutions after the vaccination. While there were no statistically significant differences among all institutions before the vaccination, there were statistically significant differences among public institutions with dramatically decreasing, university, and private clinics after the vaccination.

Similarly, a decrease in the amount of PPE was seen after vaccination. While the mean amount of PPE use was 4.69 in unvaccinated dentists, this mean was 4.33 after vaccination. It was seen that these decreases in the average amount of PPE of the participants showed a statistically significant difference. The amount of PPE regressed in dentists with the potential effect of confidence after the vaccination. Since the high transmission risk of COVID-19 and its ability to cause severe respiratory problems is well known, the existence of PPE is substantial to prevent the spreading of infection with regards to dentistry. PPE includes FFPE masks, sterilized gloves, and respirators [16]. PPE most preferred by dentists were disposable gown (85.68%), surgical mask (75.36%), N95 mask (68.63%), and FFP2/FFP3 mask (48.42%) for before vaccination, and it was detected that these percentages were 77.26%, 78.73%, 59.78%, and 39.36%, respectively, in the period after vaccination (Figure 1). In the previous study of these authors, it was detected that the rate of surgical mask use and N95/FFP2/FFP3 mask use were 81% and 47%, respectively [17]. In a study of Italian dentists, it was seen that the rates of surgical mask and FFP2/FFP3 mask use were 74.56% and 54.84%, respectively [18]. In a comprehensive study that was carried out in many countries, it was found out that the percentage of N95 mask use was 46.6% [19].

It is common knowledge that a global crisis emerged out of fear and anxiety caused by the COVID-19 outbreak. The potential quick transmission, severity, and death risk of the COVID-19 outbreak has increased psychological pressure in dentists, as well as in other healthcare professionals [20]. In a study in which participants, Italian dentists, were influenced deeply by the outbreak, 85% of participants stated fear of infection during dental procedures, and 70.2% declared that they had anxiety owing to the COVID-19 outbreak [21]. In another study, 78% of dentists stated that they had fear and anxiety in the period of the COVID-19 outbreak [22]. In the previous study of these authors, while it was detected that this rate of fear and anxiety in dentists was 68.21% [17], in the present study, it was found that the percentage of dentists with fear and anxiety before the vaccination was 76% and was seen that 35.6% of dentists stated that there was a decrease in fear and anxiety in the period after the vaccination. The consideration of leaving their institution for dentists working in a public institution due to the COVID-19 outbreak was a higher rate than dentists working in universities and private clinics, and these data were statistically significantly different. The first reason for the high percentage of dentists working in public institutions could fall within filiation teams that browsed patients with COVID-19 contact (decision taken by the Ministry of Health of the Republic of Turkey to combat the outbreak), and secondly it could be that they worked at a high pace in the period of the outbreak’s peak. Younger dentists stated that they had higher fear and anxiety due to the COVID-19 outbreak than more experienced dentists, and this difference was statistically significantly different. Among possible reasons of less fear and anxiety in dentists with 21 or more years of experience could be low patient intake and perhaps less possibility of risk of being infected, as younger dentists tend to work in filiation teams. A similar situation can also be seen in the rate of answers given to the question “Did anxiety and fear levels decrease after vaccination?”. While younger dentists fought on the frontline in the COVID-19 outbreak, they stated that their fear and anxiety decreased due to possible confidence after the vaccination, and this difference was statistically significant.

Vaccination is crucial to prevent the spreading of fatal diseases. The vaccine rejection of individuals and the existence of vaccine hesitancy reduce the rate of vaccination [23]. In this study, the participants were asked about the vaccine situation, and 98.3% of participants stated that they took the vaccine. Only eight dentists were un-vaccinated. The willingness of taking the vaccine was evaluated in Taiwanese healthcare workers by Kukreti et al. A total of 23.4% of participants stated low willingness. They speculated that this low rate was due to more control of the COVID-19 outbreak in Taiwan [24]. In another study, on the eve of sharing with the public the COVID-19 vaccines, it was stated that 28.4% of healthcare workers serving in France and Francophone countries such as Belgium and Canada showed low willingness [25]. In a cross-sectional study conducted by Fisher et al., 10% of participants communicated a negative opinion about the COVID-19 vaccine in April 2020 [26]. Di Giuseppe et al. notified that the willingness regarding COVID-19 vaccines was more in those who agreed that COVID-19 is a severe disease and those who followed scientific journals [27]. Although almost the whole of the population of participants were administered the vaccination, the vaccine confidence was 5.97 points out of 10. Among the reasons for this low confidence in the vaccine may be the risks, safety, and side effects of the vaccine, as shown in a study conducted in Hong Kong [28]. Unfortunately, a comparative study could not be conducted because no other vaccine was available in the country at the time healthcare workers were included in the mass vaccination program in Turkey. Although dentists working in universities as opposed to in private clinics and public institutions, specialist dentists as opposed to general dentists, and younger dentists as opposed to older dentists had higher vaccine confidence points, these differences were not statistically significantly different. A reason for the higher vaccine confidence points in these dentists may be the following of current publications.

A total of 14.9% of participants were diagnosed as COVID-19-positive before the vaccination, and 6.73 of were also diagnosed as COVID-19-positive after the vaccination. Only three dentists were diagnosed as COVID-19-positive pre- and post-vaccination. According to the Turkish Dental Association, 32 dentists have died since the beginning of the outbreak [29]. In a study on fear and preventive behaviors caused by the COVID-19 outbreak in dentists, 3.9% of dentists who participated in the study confirmed that they were COVID-19-positive [30]. In a study conducted by Sebastian et al. [31], the positive rate of dentists was 4%, and in the study of Alajmi et al. [32], this rate was also 10%. In a previous study conducted by these authors, the rate of dentists diagnosed with COVID-19 was 1.7% [17]. In the study of Iurcov et al., it was thought that the positive rate was low because the participants did not take patients [30]. In the current study, the number of dentists who stated that they did not receive patients from the start of the outbreak until the time of this study was only five.

It is obvious that the current study has some general limitations regarding the sample size, homogeneity of the data collected, demographic distribution, and questionnaire-based optimization needs such as the source of information obtained. However, despite these limitations, we think that the results expressed herein showed the effects of COVID-19 vaccination on the clinical behavior of dental practitioners.

## 5. Conclusions

The current study has clearly shown that vaccination has a positive effect on decreasing the anxiety levels of dental professionals and significantly improves the duration of the service. However, protective measurements should further be the main concern, regardless of the vaccination status of both the dental professional and the patient.

## Figures and Tables

**Figure 1 ijerph-18-10373-f001:**
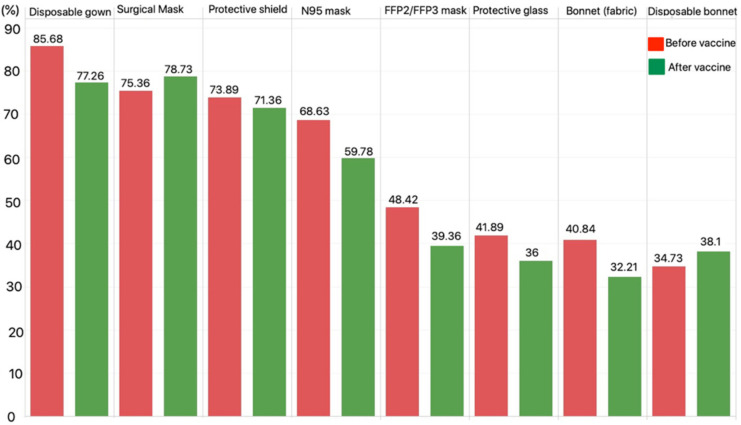
The usage of personal protective equipment in the period before and after the vaccination.

**Table 1 ijerph-18-10373-t001:** Questions of the survey.

Age (… year(s))
Gender (W/M)
Specialty (if any)
Experience duration (… year(s))
Which institution?
Which city?
Any systemic disease? (Y/N)
Have you been given the COVID-19 vaccine? (Y/N)
What date were you vaccinated?
Did you have antibody measurement after vaccination? If you have, please indicate the value and which of the Spike or Neutralizing antibody tests were performed? (Y/N)
What is your confidence value in the vaccine administered to you? (score from 0–10; 0: none, 10: I trust)
Before the vaccination, what number of patients did you examine? (per day)
After the vaccination, what number of patients did you examine? (per day)
Before the vaccination, did you utilize personal protective equipment in the dental examination? (surgical mask, N95, FFP2/3, etc.)
After the vaccination, did you utilize personal protective equipment in the dental examination? (surgical mask, N95, FFP2/3, etc.)
Had you been COVID-19-positive before the vaccination? (Y/N)
Have you been COVID-19 positive after the vaccination? (Y/N)
Before the vaccination, did you use antiseptic solutions before the dental examination? (Y/N)
After the vaccination, did you use antiseptic solutions before the dental examination? (Y/N)
Did you consider leaving institutions before the vaccination? (Y/N)
Did you have decreased fear and anxiety after the vaccination? (Y/N)
Did you have fear and anxiety due to the COVID-19 pandemic? (Y/N)

(Y: yes, N: no, W: woman, M: man).

**Table 2 ijerph-18-10373-t002:** The distribution of COVID-19 and sociodemographic data of participants.

		*n*	%
Gender	Men	196	41.3
	Women	279	58.7
Institutions	Public institution	174	36.6
	Private clinics	133	28
	Universities	168	35.4
Experience duration	0 and 10 years	184	38.7
	11 and 20 years	194	40.9
	Over 21 years	97	20.4
Specialty status	Specialist dentists	146	30.7
	General dentists	329	69.3
Systemic diseases	Yes	48	10.1
	No	427	89.9
Vaccination	Yes	467	98.3
	No	8	1.7
Antibody measurements	Yes	96	20.5
	No	379	79.5
COVID (+) before the vaccine	Yes	71	14.9
	No	404	85.1
COVID (+) after vaccine	Yes	32	6.7
	No	443	93.3
Preoperative mouth rinsing before the vaccine	Yes	256	53.9
	No	219	46.1
Preoperative mouth rinsing after vaccine	Yes	199	41.9
	No	276	58.1
The amount of PPE before vaccine (mean)		4.69	
The amount of PPE after vaccine (mean)		4.33	

**Table 3 ijerph-18-10373-t003:** Alterations before and after vaccination for all participants.

	before Vaccine (*n*)	after Vaccine (*n*)	*p*-Value
Examined number of patients (per day)	0–10: 396	0–10: 366	**0.039 ***
	11–20: 49	11–20: 71	
	21+: 30	21+: 38	
PPE use during the dental examination (mean)	4.69	4.32	**0.001 ***
COVID (+) diagnosed	71	32	**0.001 ***
The use of preoperative mouth rinsing	256	199	**0.001 ***

* Significant parameters: 0–10: number of patients examined between 0 and 10 per day, 11–20: number of patients examined between 11 and 20 per day, 21+: number of patients examined daily was 21 or more; PPE: personal protective equipment. Bold numbers: statistically significant differences.

**Table 4 ijerph-18-10373-t004:** Alterations in before and after vaccination according to institutions, specialty status, and experience durations.

	Public Institutions	Private Clinics	Universities	General Dentists	Specialist Dentists	0–10 Years Experience	11–20 Years Experience	21 Years and Over Experience
The number of examined patients per day before vaccination (*n*)	0–10: 145	0–10: 115	0–10: 136	0–10: 123	0–10: 273	0–10: 153	0–10: 163	0–10: 80
	11–20: 19	11–20: 10	11–20: 20	11–20: 14	11–20: 35	11–20: 18	11–20: 16	11–20: 15
	21+: 10	21+: 8	21+: 12	21+: 9	21+: 21	21+: 13	21+: 15	21+: 2
The number of examined patients per day after vaccination (*n*)	0–10: 134	0–10: 106	0–10: 126	0–10: 115	0–10: 251	0–10: 136	0–10: 160	0–10: 70
	11–20: 25	11–20: 18	11–20: 27	11–20: 20	11–20: 51	11–20: 29	11–20: 19	11–20: 23
	21+: 15	21+: 8	21+: 15	21+: 11	21+: 27	21+: 19	21+: 15	21+: 4
	*p* = 0.16	*p* = 0.354	*p* = 0.216	*p* = 0.304	*p* = 0.072	*p* = 0.031 *	*p* = 0.876	*p* = 0.089
Personal protective equipment used in dental examination before the vaccine	4.79	4.6	4.66	4.67	4.7	4.52	4.9	4.59
Personal protective equipment used in dental examination after vaccine	4.54	4.15	4.24	4.52	4.24	4.21	4.56	4.06
	*p* = 0.088	*p* = **0.006 ***	*p* = **0.008 ***	*p* = 0.323	*p* = **0.001 ***	*p* = **0.022 ***	*p* = **0.022 ***	*p* = **0.012 ***
COVID (+) diagnosed before vaccine (*n*)	29	21	21	20	51	18	29	24
COVID (+) diagnosed after vaccine (*n*)	17	7	8	11	21	16	9	7
	*p* = 0.09	*p* = **0.011 ***	*p* = **0.026 ***	*p* = 0.137	*p* = **0.001 ***	*p* = 0.864	*p* = **0.002 ***	*p* = **0.002 ***
The use of preoperative mouth rinsing before the vaccine	Y: 87	Y: 74	Y: 95	Y: 80	Y: 176	Y: 89	Y: 113	Y: 54
	N: 87	N: 49	N: 73	N: 66	N: 153	N: 95	N: 81	N: 43
The use of preoperative mouth rinsing after vaccine	Y: 55	Y: 66	Y: 78	Y: 60	Y: 139	Y: 86	Y: 73	Y: 40
	N: 119	N: 67	N: 90	N: 86	N: 190	N: 98	N: 121	N: 57
	*p* = **0.001 ***	*p* = 0.366	*p* = 0.068	*p* = **0.016 ***	*p* = **0.003 ***	*p* = 0.82	*p* = **0.001 ***	*p* = 0.066

* Significant parameters: Y: yes, N: no, 0–10: number of patients examined between 0 and 10 per day, 11–20: number of patients examined between 11 and 20 per day, 21+: number of patients examined daily was 21 or more; PPE: personal protective equipment. Bold numbers: statistically significant differences.

**Table 5 ijerph-18-10373-t005:** The evaluation according to institutions.

	Public Institutions	Private Clinics	Universities	*p*-Value
The vaccine confidence interval value	5.88	6	6.05	0.763
Considering leaving institutions before the vaccination (*n*/%)	Y: 42/24.1%	Y: 24/18.1%	Y: 24/14.3%	**0.024 ***
	N: 132/75.9%	N: 109/81.9%	N: 144/85.7%	
Decreasing of fear and anxiety after the vaccination (*n*/%)	Y: 69/39.6%	Y: 39/29.3%	Y: 61/36.3%	0.43
	N: 105/60.4%	N: 94/70.7%	N: 107/63.7%	
Fear and anxiety due to COVID-19 pandemic (*n*/%)	Y: 134/77%	Y: 103/77.4%	Y: 124/75.6%	0.462
	N: 40/23%	N: 30/22.6%	N: 44/21.4%	
Preoperative mouth rinsing before vaccine	Y: 87/50%	Y: 74/55.6%	Y: 95/56.5%	0.179
	N: 87/50%	N: 49/44.4%	N: 73/45.5%	
Preoperative mouth rinsing after vaccine	Y: 55/31.6%	Y: 66/49.6%	Y: 78/46.4%	**0.006 ***
	N: 121/68.4%	N: 67/50.4%	N: 90/53.6%	
The number of examined patients per day before vaccination (*n*)	0–10: 145	0–10: 115	0–10: 136	0.627
	11–20: 19	11–20: 10	11–20: 20	
	21+: 10	21+: 8	21+: 12	
The number of examined patients per day after vaccination (*n*)	0–10: 134	0–10: 106	0–10: 126	0.854
	11–20: 25	11–20: 18	11–20: 27	
	21+: 15	21+: 8	21+: 15	
The amount of PPE before vaccine (mean)	4.79	4.6	4.66	
The amount of PPE after vaccine (mean)	4.54	4.15	4.24	

* Significant parameters: Y: yes, N: no, 0–10: number of patients examined between 0 and 10 per day, 11–20: number of patients examined between 11 and 20 per day, 21+: number of patients examined daily was 21 or more; PPE: personal protective equipment. Bold numbers: statistically significant differences.

**Table 6 ijerph-18-10373-t006:** The evaluation according to specialty status.

	Specialist Dentists	General Dentists	*p*-Value
The vaccine confidence interval value	6.03	5.85	0.414
Considering leaving institutions before the vaccination (*n*/%)	Y: 59/17.9%	Y: 32/21.9%	0.565
	N: 270/82.1%	N: 114/78.1%	
Decreasing of fear and anxiety after the vaccination (*n*/%)	Y: 120/36.5%	Y: 49/33.5%	0.744
	N: 209/63.5%	N: 97/66.5%	
Fear and anxiety due to COVID-19 pandemic (*n*/%)	Y: 240/72.9%	Y: 124/84.9%	**0.007 ***
	N: 89/27.1%	N: 22/15.1%	
Preoperative mouth rinsing before vaccine	Y: 176/53.5%	Y: 80/54.8%	0.627
	N: 153/46.5%	N: 66/45.2%	
Preoperative mouth rinsing after vaccine	Y: 139/42.2%	Y: 60/41.1%	0.972
	N: 190/57.8%	N: 86/58.9%	
The number of examined patients per day before vaccination (*n*)	0–10: 273	0–10: 123	0.808
	11–20: 35	11–20: 14	
	21+: 21	21+: 9	
The number of examined patients per day after vaccination (*n*)	0–10: 251	0–10: 115	0.578
	11–20: 51	11–20: 20	
	21+: 27	21+: 11	
The amount of PPE before vaccine (mean)	4.7	4.67	
The amount of PPE after vaccine (mean)	4.24	4.52	

* Significant parameters: Y: yes, N: no, 0–10: number of patients examined between 0 and 10 per day, 11–20: number of patients examined between 11 and 20 per day, 21+: number of patients examined daily was 21 or more; PPE: personal protective equipment. Bold numbers: statistically significant differences.

**Table 7 ijerph-18-10373-t007:** The evaluation according to experience duration.

	0–10 Years- Experience	11–20 Years- Experience	21 Years and Over Experience	*p*-Value
The vaccine confidence interval value	6.13	5.9	5.85	0.475
Considering leaving institutions before the vaccination (*n*/%)	Y: 36/19.6%	Y: 41/21.1%	Y: 14/14.4%	0.534
	N: 148/80.4%	N: 153/78.9%	N: 83/85.6%	
Decreasing of fear and anxiety after the vaccination (*n*/%)	Y: 71/38.6%	Y: 74/38.1%	Y: 30/30.9%	**0.046 ***
	N: 113/61.4%	N: 120/61.9%	N: 67/69.1%	
Fear and anxiety due to COVID-19 pandemic (*n*/%)	Y: 140/76.1%	Y: 158/81.4%	Y: 66/68.1%	**0.029 ***
	N: 44/23.9%	N: 36/18.6%	N: 31/31.9%	
Preoperative mouth rinsing before vaccine	Y: 89/48.4%	Y: 113/58.2%	Y: 54/55.7%	0.134
	N: 95/51.6%	N: 81/41.8%	N: 43/44.3%	
Preoperative mouth rinsing after vaccine	Y: 86/46.7%	Y: 73/37.6%	Y: 40/41.2%	0.255
	N: 98/53.3%	N: 121/62.4%	N: 57/58.8%	
The number of examined patients per day before vaccination (*n*)	0–10: 153	0–10: 163	0–10: 80	0.795
	11–20: 18	11–20: 16	11–20: 15	
	21+: 13	21+: 15	21+: 2	
The number of examined patients per day after vaccination (*n*)	0–10: 136	0–10: 160	0–10: 70	**0.041 ***
	11–20: 29	11–20: 19	11–20: 23	
	21+: 19	21+: 15	21+: 4	
The amount of PPE before vaccine (mean)	4.52	4.9	4.59	
The amount of PPE after vaccine (mean)	4.21	4.56	4.06	

* Significant parameters: Y: yes, N: no, 0–10: number of patients examined between 0 and 10 per day, 11–20: number of patients examined between 11 and 20 per day, 21+: number of patients examined daily was 21 or more; PPE: personal protective equipment. Bold numbers: statistically significant differences.

## Data Availability

Not applicable.

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
