# Peer review of "The Impact of COVID-19 Vaccination on Anxiety Levels of Turkish Dental Professionals and Their Attitude in Clinical Care: A Cross-Sectional Study"

_ijerph, 2021, doi:10.3390/ijerph181910373_

Round 1

Reviewer 1 Report

Line 58 English language needs revising: "Dentistry is taken into account among the risky professions in the period of the"

Line 76 It is not possible to understand the meaning of "This study was performed with the evaluation of dentists

Lines 83-83 very hard to understand English ", and they were become obligatory to approve the form to participate in the study during the stage of respondence the online questionnaire. "

Line 150 "significantly different (p = 0.09)" (statistically significant difference)

Line 156 "but the decreases in general dentists were not statistically significant differences" - please correct, the meaning is not straightforward.

Lines 161-162 " It was learned that the number of covid (+) diagnosed de- creased in all groups after vaccination" - grammatically clumsy and not suitable for Results, rather should go to discussion if it is author's subjective interpretation of the results.

Table 4, what is "interventional patients" - I am afraid this is not the right term.

Line 479 "vaccination has a positive effect on the anxiety levels of dental professionals " I would change "vaccination has a positive effect on decreasing the anxiety levels of dental professionals "

Author Response

Dear reviewer,

Corrections have been made in the sentences and lines you have pointed out and have been marked with yellow color. On behalf of the authors, I thank you for your suggestions.

Best regards

Reviewer 2 Report

I really think that this is an excellent paper. I have only very few comments:

All tables need some clarification and modification (i.e. in Table 3 what the asterisk stay for?)

Please be consistent for all the manuscript about the number of decimals.

Please remove in the discussion section all numerical data, there is no need of that.

Author Response

Dear reviewer,

In the sentences and lines you have pointed out, the corrections you have made have been made and marked with red color. On behalf of the authors, I thank you for your suggestions.

Best regards

Reviewer 3 Report

Thank you for giving me this opportunity to review the article entitled, "The Impact of COVID-19 Vaccination on Anxiety Levels of Turkish Dental Professionals and Their Attitude in Clinical Care: A Cross-Sectional Study ". The article covers an important topic these days, demonstrating the importance of vaccination in this and many other pathologies.

  • Despite this, the article is very exhausted. In the "results" there is a very detailed description, which will repeat the information in the tables. There should be an abbreviation of that section for the reader to get the article, describing only the most important results.

Author Response

Dear reviewer,

First of all, on behalf of the authors, thank you for your nice comments. According to your recommendations we have excluded the data which were statistically insignificant to improve clarity the of the paper. 

Best regards